# Effect of Point Defects on Electronic Structure of Monolayer GeS

**DOI:** 10.3390/nano11112960

**Published:** 2021-11-04

**Authors:** Hyeong-Kyu Choi, Janghwan Cha, Chang-Gyu Choi, Junghwan Kim, Suklyun Hong

**Affiliations:** Department of Physics, Graphene Research Institute, and GRI-TPC International Research Center, Sejong University, Seoul 05006, Korea; chk75541@gmail.com (H.-K.C.); heatpoint@nate.com (J.C.); power_ccq@naver.com (C.-G.C.); uprightshine0415@naver.com (J.K.)

**Keywords:** germanium monosulfide (GeS), defect, formation energy, vacancy healing, electronic structure, piezoelectric coefficients, DFT calculations, two-dimensional materials

## Abstract

Using density functional theory calculations, atomic and electronic structure of defects in monolayer GeS were investigated by focusing on the effects of vacancies and substitutional atoms. We chose group IV or chalcogen elements as substitutional ones, which substitute for Ge or S in GeS. It was found that the bandgap of GeS with substitutional atoms is close to that of pristine GeS, while the bandgap of GeS with Ge or S vacancies was smaller than that of pristine GeS. In terms of formation energy, monolayer GeS with Ge vacancies is more stable than that with S vacancies, and notably GeS with Ge substituted with Sn is most favorable within the range of chemical potential considered. Defects affect the piezoelectric properties depending on vacancies or substitutional atoms. Especially, GeS with substitutional atoms has almost the same piezoelectric stress coefficients $e_{ij}$ as pristine GeS while having lower piezoelectric strain coefficients $d_{ij}\;$ but still much higher than other 2D materials. It is therefore concluded that Sn can effectively heal Ge vacancy in GeS, keeping high piezoelectric strain coefficients.

## 1. Introduction

Since theoretically proposed graphene [1] was experimentally separated from graphite [2], the research on graphene has been actively conducted because of its fast charge mobility [3,4], good thermal conductivity [3,5], and so on. Such interest has expanded to a variety of two-dimensional materials with sizable bandgap and outstanding electronic properties, such as transition metal dichalcogenides (TMDs) [6,7,8,9,10,11,12,13,14,15,16] and black phosphorus (BP) [17,18,19,20]. Since BP has been considered as a promising material for applications, group IV monochalcogenides with a similar structure to BP, such as GeS, GeSe, SnS, and SnSe, have also attracted attention. Interestingly, group IV monochalcogenides are more resistant to oxidation than BP [21]. These materials, such as group IV monochalcogenides, can be used in photovoltaics [22,23] and thermoelectrics [24,25,26,27,28,29,30,31]. In addition, the problem of efficient energy storage [32,33,34,35,36] and very strong piezoelectric effect [37,38,39] have received much attention. Especially, multi-layered GeS photodetectors exhibit device robustness, photoswitching stability, and long-term durability [40]. Among MX (M = Ge, Sn; X = S, Se) sheets, GeS sheet has the lowest activation barriers and holds Li atoms strongly with the highest adsorption energies; at the same time, low, open-circuit voltage and higher capacity can make it the best choice for Li-ion batteries [32,41]. The extraordinary sensitivity and selectivity of GeS monolayer for ${\mathrm{NO}}_{2}$ molecules suggest that GeS monolayer is a potential sensing material for ${\mathrm{NO}}_{2}$ gas [42]. Moreover, GeS monolayers are proposed as efficient photocatalysts for water splitting [43,44,45].

In general, defects play an important role in determining electronic properties. In 2D materials, defects, such as edges, grain boundaries, strain, vacancies, substitutional atoms, and dopants, can lead to localized electronic properties, which is different from the case of pristine 2D materials. These properties are sometimes beneficial ones or sometimes harmful ones that should be removed. For example, mechanically isolated MoS_2_ shows better mobility than chemically deposited MoS_2_ [46,47]. Furthermore, MoS_2_ samples obtained using chemical vapor deposition, physical vapor deposition, and mechanical delamination have about ten times lower mobility than theoretical expectations because of the presence of grain boundaries or point defects [48]. Such defects can cause optical properties to change. In MoS2, the shift of the first-order Raman mode is affected by the concentration of S vacancies [49]. Moreover, intensive calculations related to defects and modification of 2D materials have been reported [50,51,52,53,54,55,56].

On the other hand, the defects of group IV monochalcogenides have been actively studied. For example, group IV monochalcogenides are more resistant to oxidation than BP, which have the same structure with group IV monochalcogenides [21]. It was predicted by simulation studies that the original bandgap can be recovered by substituting chalcogen vacancies with oxygen. In terms of formation energy, Ge vacancies are more likely to form than S vacancies, but there has been little discussion on how to heal Ge vacancies [21]. Therefore, further discussion is needed on how to effectively reduce and heal vacancies in group IV monochalcogenides.

In this study, we investigate structural and electronic properties related to point defects, such as Ge and S vacancies, in GeS that is one of group IV monochalcogenides. Compared to pristine GeS, we focus on the change caused by vacancies and substitutional atoms. Group IV elements, such C, Si, and Sn, are chosen as the substitutional atom of Ge, while chalcogen elements, such as Se and Te, are chosen as that of S. Specifically, the focus is on the case where the properties of defective GeS are restored by substitutional atoms.

## 2. Calculation Method

To understand the structural and electronic properties of group IV monochalcogenides, we employed a first-principles approach based on the spin-polarized density functional theory [57,58], as implemented in the Vienna ab-initio simulation package (VASP) [59,60], which is used for calculations of density of states (DOS) and elastic and piezoelectric coefficients. Note that spin-polarized calculations lead to no spin polarization for all the systems considered. The core and valence electrons were treated with the projector-augmented wave (PAW) [61] method. The exchange correlation energy is described by the generalized gradient approximation (GGA) using the Perdew-Burke-Ernzerhof (PBE) [62] functional. The Kohn-Sham orbitals are expanded in a plane-wave basis with a cutoff energy of 400 eV. To model the pristine and defective GeS, the 3 × 3 supercells are periodic in the monolayer plane, and large vacuum regions (12 Å) are included to impose periodic boundary conditions in the perpendicular direction. The Brillouin zone (BZ) is sampled using a Γ-centered 14 × 10 × 1 grid for the monolayer, following the scheme proposed by Monkhorst-Pack [63]. The convergence criteria for electronic and ionic relaxations are $10^{-6}$ eV and $10^{-3}$ eV/Å, respectively. The charge transfers are calculated with decomposition of charge density into atomic contributions by using the Bader charge analysis method [64]. To obtain the piezoelectric properties, the elastic coefficient $C_{ij}$ is calculated by using strain–stress relationship (SSR) [65], and the piezoelectric stress coefficients $e_{ij}$ are calculated by the density functional perturbation theory (DFPT) method [66,67] by the VASP code.

## 3. Results and Discussion

### 3.1. Crystal Structure and Energetics

The pristine GeS is one of group IV monochalcogenide that has an orthorhombic structure. Bulk GeS forms a $Pnma-D_{2h}^{16}\;$ space group and shows a lower symmetry than bulk BP ($Pnma-D_{2h}^{18}$) because it is composed of two kinds of atoms, unlike BP. If the dimension is reduced to monolayer, the space group changes to $Pmn2_{1}-C_{2v}^{7}$. Such difference in symmetry affects the piezoelectric coefficient, which will be discussed below in Section 3.3. The in-plane lattice constants *a* = 4.54 Å, *b* = 3.63 Å are used as shown in Figure 1. In this study, we use 3 × 3 supercell including about 12 Å vacuum.

Vacancies or substitutional atoms are introduced as defects, of which structures are shown in Figure 2. Vacancies can occur at either S or Ge position. For S vacancy ($V_{S}$), the Ge atoms nearest to the S vacancy site gather inside the GeS layer, while for Ge vacancy ($V_{\mathrm{Ge}}$), the S atoms nearest to the Ge vacancy site move away. When foreign atoms are introduced into vacancy sites, the structural changes occur: the substituting atoms for the vacancy positions lie in the GeS plane or protrude from the plane, depending on their sizes. If C atoms substitute for Ge atoms ($C_{\mathrm{Ge}}$), the C atoms lie in the plane of GeS layer, while if Sn atoms replaces Ge atoms (${\mathrm{Sn}}_{\mathrm{Ge}}$), the Sn atoms are placed slightly above the GeS plane. For the same reason, if Te is placed in the S vacancy site (${\mathrm{Te}}_{S}$), it is placed slightly above the GeS plane. Note that in the case of $C_{\mathrm{Ge}}$, the C atom binds to the surrounding S atoms, and thus, the S atoms move too far towards C, breaking its bonds with Ge atoms.

To find out the stability of monolayer GeS with defects, we consider the formation energy of the system with defects. The formation energy $E_{formation}$ is defined as follows:(1)$$E_{formation}=E_{defect}-\left(E_{pristine}+\sum n_{i}\mu_{i}\right)$$
where *E_pristine_* is the total energy of pristine GeS, and $E_{defect}$ is that of monolayer GeS with defects (vacancies or substitutional atoms) at the Ge or S position. Here, $n_{i}$ is the number of the i element that has been removed or introduced for substitution, and $\mu_{i}$ is the corresponding chemical potential. Since only one element is removed or substitutes in the 3 × 3 supercell, the value of $n_{i}$ is −1 or +1. To measure the chemical potential, we used the relation that $\mu_{\mathrm{GeS}}≅\mu_{\mathrm{Ge}}$ + $\mu_{S}$, where $\mu_{\mathrm{GeS}}$ means the total energy per formula unit in pristine GeS. Since the systems under study are assumed to be in chemical and thermal equilibrium with bulk, one may be able to use the bulk energy, i.e., the total energy per atom of the specific bulk crystal, as the chemical potential [68,69,70]. That is, the chemical potentials $\mu_{i}$ for substitutional atoms are calculated from their bulk structures. In the Ge-rich environment, $\mu_{\mathrm{Ge}}$ is the total energy per atom in the diamond-structured solid Ge, which enables one to calculate $\mu_{S}$ as $\mu_{\mathrm{GeS}}$ − $\mu_{\mathrm{Ge}}$. On the other hand, in an S-rich environment, $\mu_{S}$ is the total energy per atom in α-S8 crystal, and thus, $\mu_{\mathrm{Ge}}$ is calculated as $\mu_{\mathrm{GeS}}$ − $\mu_{S}$. The chemical potentials of chalcogen elements, such as Te and Se, are calculated from the crystal structures of $P3_{1}21$ Te [71] and hexagonal Se [72,73,74], respectively.

The results of the formation energies as a function of chemical potential are represented in Figure 3. Specifically, the formation energies are calculated at two points, such as Ge-rich and S-rich environments for each line. For the case of vacancies, it is natural that $V_{S}$ has difficulty in creating in S-rich environment, while $V_{\mathrm{Ge}}$ has difficulty making a Ge-rich environment. The formation energy of $V_{\mathrm{Ge}}$ is smaller than that of $V_{S}$ except for the chemical potential range around Ge-rich environment, so it is considered that $V_{\mathrm{Ge}}$ is more likely to be found than $V_{S}$ in relatively wider range of chemical potential considered. Indeed, it is experimentally expected that $V_{\mathrm{Ge}}$ is more frequently observed than $V_{S}$. A remarkable point is that the formation energy of ${\mathrm{Sn}}_{\mathrm{Ge}}$ is negative for any value of chemical potential between in S-rich and Ge-rich environments, which implies that ${\mathrm{Sn}}_{\mathrm{Ge}}$ is energetically favorable and forms spontaneously in the presence of Ge vacancies and Sn atoms. Such Sn substitution may play an important role in effective vacancy healing. In addition, ${\mathrm{Si}}_{\mathrm{Ge}}$ has negative formation energy in an S-rich environment. In contrast, the formation energies of $C_{\mathrm{Ge}}$ are relatively large, positive values in the chemical potential range considered, leading to its instability.

### 3.2. Electronic Properties

To study the electronic structure of defective GeS, we calculated density of states (DOS) of GeS. Figure 4 shows total DOS and partial DOS (PDOS) for (a) pristine GeS and defective GeS with (b) $V_{S}$; (c) ${\mathrm{Se}}_{S}$ and ${\mathrm{Te}}_{S}$; (d) $V_{\mathrm{Ge}}$; and (e) $C_{\mathrm{Ge}}$, ${\mathrm{Si}}_{\mathrm{Ge}}$, and ${\mathrm{Sn}}_{\mathrm{Ge}}$. Here, the black line represents total DOS, while colored lines represent PDOS corresponding to the same-colored atoms drawn in dark in the inset. The bandgap of pristine GeS is 1.76 eV, which is listed in Table 1 along with the bandgaps of defective GeS. When vacancies are introduced, the bandgaps become smaller for the cases of both $V_{\mathrm{Ge}}$ and $V_{S}$, of which bandgaps are 0.04 eV and 1.04 eV, respectively. Defects such as $V_{\mathrm{Ge}}$ and $V_{S}$ present vacancy states inside the forbidden band region of pristine GeS, leading to the doping effect. This agrees well with previous research results [21]. Refer to Appendix A for band structures of pristine and defective GeS.

For the case of $V_{S}$, the vacancy state in PDOS of $V_{S}$ is located just below the Fermi energy, which makes the bandgap become narrower and act as a donor state. The vacancy state is created by adjacent Ge atoms surrounding S vacancy. The contribution to the vacancy state by the Ge atoms amounts to the most portion of vacancy state. Structurally, when $V_{S}$ forms, three surrounding Ges gather to be close each other and thus contribute to the formation of the vacancy state, while atoms farther away from S vacancy do not contribute the vacancy state. In contrast, in the case of $V_{\mathrm{Ge}}$, a relatively wider vacancy state appears just above the Fermi energy acting as an acceptor state. In addition, it is difficult to say that the states just below the Fermi energy are made only by the contributions of adjacent atoms.

When the substitutional atom is introduced for either Ge site or S site, the bandgap becomes larger than those of both cases of $V_{\mathrm{Ge}}$ and $V_{S}$ and has a value close (within ~10%) to that of pristine GeS, as shown in Table 1. When S is substituted with Se or Te, the structural properties are almost the same as that of pristine GeS because S, Se, and Te belong to the same group, which makes the electronic properties, such as bandgap value, almost similar. Similarly, when Ge is substituted with Si and Sn, it is likewise the case. However, in case of $C_{\mathrm{Ge}}$, as mentioned earlier in the structural description, the S atoms around the substituted C atom become closer to the C atom and further away from outer Ge atoms, so $C_{\mathrm{Ge}}$ may have an electronic structure similar to $V_{S}$. Especially, considering the formation energy discussed above, ${\mathrm{Sn}}_{\mathrm{Ge}}$ is energetically most stable among the substitutional cases, so when Ge is substituted with Sn to replace Ge in GeS, the original electronic properties of pristine GeS are almost recovered. 

It is worthwhile to ask whether the band gaps of pristine and defective GeS are direct or indirect. Pristine monolayer GeS has an indirect band gap, which has been already reported in many previous studies [75,76,77]. Comparison of PDOS profiles between pristine and substitutional GeS except $C_{\mathrm{Ge}}$ shows similarities to each other, which makes one expect that substitutional GeS except $C_{\mathrm{Ge}}$ have indirect band gaps like the pristine case. This can be confirmed by the band structure plots shown in Appendix A. For $V_{\mathrm{Ge}}$, $V_{S}$ and $C_{\mathrm{Ge}}$, the PDOS profiles are much changed from that of the pristine case. For detailed band structures, including the locations of the valence band minimum (VBM) and conduction band minimum (CBM), see Appendix A: the band gaps of $V_{\mathrm{Ge}}$, $V_{S}$ and $C_{\mathrm{Ge}}$ are indirect.

Based on the Bader charge analysis, we found charge transfers between the substitutional atom and neighboring atoms. For Ge-substituted cases, Si loses 0.36*e* to neighboring S atoms, while C and Sn gain 2.55*e* and 0.11*e* from them, respectively. Note that a relatively large gain occurs for C due to large deformation around C. For S-substituted cases, Se and Te lose 1.28*e* and 1.62*e* to neighboring Ge atoms, respectively. In the cases of Ge or S vacancies, the S atoms nearest to the Ge vacancy site lose 0.25*e* per S, while the Ge atoms nearest to S vacancy site gain 0.62*e* per Ge compared to the pristine case.

### 3.3. Piezoelectric Properties

Since the piezoelectric property is a ground-state one, DFT is a useful tool for predicting those properties, and a great deal of previous studies on GaN [78] and MoS2 [79,80] show high agreement between DFT and experiments. As mentioned earlier, monolayer GeS belongs to monolayer group IV monochalcogenides, which belongs to $Pmn2_{1}-C_{2v}^{7}$ space group. Because of the prediction that this group has strong piezoelectric properties using DFT [37,38], the group has received a great deal of attention and has been followed by related subsequent studies. Specifically, there have been attempts to maximize the piezoelectric effects by slightly modifying the structure of group IV monochalcogenides, but the piezoelectric effect has not been greatly increased [39]. Here, we investigated how defects influence the piezoelectric properties of GeS depending on vacancies or substitutional atoms. In particular, based on the formation energy results, the more stable cases (${\mathrm{Sn}}_{\mathrm{Ge}}$, ${\mathrm{Se}}_{S}$) among the substitutions of Ge or S with foreign atoms in GeS are considered.

To know the piezoelectric properties, we calculated the elastic coefficients $C_{ij}$ and further piezoelectric coefficients by using VASP [65,66,67]. The elastic energy Δ*u* is defined by
(2)$$\Delta u\sum_{i=1}^{6}\sum_{j=1}^{6}\frac{1}{2}C_{ij}\varepsilon_{i}\varepsilon_{j}$$
where $C_{ij}$ is the elastic coefficient, and $\varepsilon_{i}$ is the strain in the i direction in Voigt notation [81]. Since we dealt with 2D monolayer GeS, the elastic energy Δ*u* is given as energy per unit area. The number of 36 components in Δ*u* can be reduced by considering the structural symmetry. Orthorhombic structure needs just nine independent elements [82]$:\;C_{11}$, $C_{22}$, $C_{33}$, $C_{44}$, $C_{55}$, $C_{66}$, $C_{12}$, $C_{13}$, and $C_{23}$. For monolayer GeS, only in-plane directions are considered with the restriction that angles between lattice vectors do not change. Then, only $C_{11}$, $C_{22}$, and $C_{12}$ need to be considered among nine elastic coefficients. The elastic energy expression is now reduced to
(3)$$\Delta u=\frac{1}{2}C_{11}\varepsilon_{1}^{2}+\frac{1}{2}C_{22}\varepsilon_{2}^{2}+C_{12}\varepsilon_{1}\varepsilon_{2},$$

The calculation results for elastic coefficients are listed in Table 2, where only the results for the relaxed-ion case that can be compared with the experiment are given along with the values from previous studies for pristine GeS and ${\mathrm{MoS}}_{2}$ for comparison. It was found that the present results agree well with the previous theoretical calculations.

Next, we calculated piezoelectric stress coefficients $e_{ij}$, showing how polarization changes with strain. The piezoelectric stress coefficients $e_{ij}$ are given by
(4)$$e_{ij}=\frac{\partial P_{i}}{\partial \varepsilon_{j}}$$

The calculation results are listed in Table 3. Note that $V_{\mathrm{Ge}}$ and $V_{S}$ have different piezoelectric stress coefficients from each other. $V_{\mathrm{Ge}}$ has piezoelectric stress coefficients $e_{11}$ and $e_{12}$ larger by 36% and 81% than those of pristine GeS, respectively. $V_{S}$ has $e_{11}$ close to (slightly smaller than) that of pristine GeS while having smaller $e_{12}$ by 20%. However, ${\mathrm{Sn}}_{\mathrm{Ge}}$ and ${\mathrm{Se}}_{S}$ have $e_{11}$ and $e_{12}$ close to those of pristine GeS, which means that they will experience similar polarization changes for similar distortions.

On the other hand, the relationship between $d_{ij}$, $C_{ij}$, and $e_{ij}$ is given by
(5)$$e_{ij}=\sum d_{ik}C_{kj}$$
where $e_{ij}$ or $d_{ij}$ are piezoelectric (stress or strain) coefficients that represent how polarization changes when strain or stress is changed, respectively. Note that both piezoelectric coefficients $e_{ij}$ and $d_{ij}$ are connected by elastic constants $C_{ij}$, showing how stress is induced with strain. Since it is sometimes convenient to use $d_{ij}$ (change of polarization depending on stress), we found $d_{ij}$ from Equation (5). The results for piezoelectric strain coefficients $d_{ij}$ are listed in Table 4. $V_{\mathrm{Ge}}$ has piezoelectric strain coefficient $d_{11}$ close (within 6%) to pristine GeS, and $V_{S}$, ${\mathrm{Sn}}_{\mathrm{Ge}}$, and ${\mathrm{Se}}_{S}$ have smaller $d_{11}$ by 17~21%, while all defective GeS’s have smaller $d_{12}$ by 19~27%. Based on these results, it was found that $V_{\mathrm{Ge}}$ has larger $e_{ij}$ and $d_{ij}$ values than $V_{S}$.

From the results listed in Table 2, Table 3 and Table 4, it is worthwhile to emphasize the relation between $d_{ij}$, $C_{ij}$, and $e_{ij}$: ${\mathrm{Sn}}_{\mathrm{Ge}}$ and ${\mathrm{Se}}_{S}$ have larger elastic constants than those of pristine GeS. Especially, for ${\mathrm{Sn}}_{\mathrm{Ge}}$ and ${\mathrm{Se}}_{S}$, the larger value of $C_{22}$ by ~10% makes them have smaller piezoelectric strain coefficients $d_{11}$ and $d_{12}$ even though they have piezoelectric stress coefficients $e_{11}$ and $e_{12}$ close to those of pristine GeS. Note that even defective GeS have larger piezoelectric strain coefficients $d_{ij}$ by one or two orders of magnitude than other bulk piezoelectric materials, which are frequently used ones, such as α-quartz [83] and wurtzite AlN [84], and recently emerging 2D materials, such as MoS_2_ [79] and GaSe [85]. Such larger piezoelectric coefficients can be used for devices applications. For better applications of piezoelectric properties of defective materials with vacancies or substitutional atoms, detailed experimental studies elucidating the relation between $d_{ij}$, $C_{ij}$, and $e_{ij}$ are needed.

## 4. Conclusions

We have studied the effects of defects on atomic and electronic structure in monolayer GeS. Vacancies in GeS create doping states, which are located just above or below the Fermi energy. Such vacancy states disappear, and the original bandgap of pristine GeS is recovered when the vacancies are substituted with foreign atoms. In terms of the formation energy, GeS with Ge replaced by Sn is found to be energetically most favorable in both S-rich and Ge-rich environments among all the defective GeS cases considered. Defective GeS has smaller piezoelectric strain coefficients $d_{ij}$ by 20–30% than pristine GeS except for $d_{11}$ of $V_{\mathrm{Ge}}$, but these values ($d_{ij}$) are much larger than those of other piezoelectric materials. Moreover, piezoelectric stress coefficients $e_{ij}$ of GeS with substitutional atoms are almost the same as those of pristine GeS. Therefore, it is concluded that substitutional atoms act as vacancy healers to help restore the properties of pristine GeS; Especially, the Sn substitution would make GeS with Ge vacancies stable and allow it to be utilized for applications in sensitive devices.

## Figures and Tables

**Figure 1 nanomaterials-11-02960-f001:**
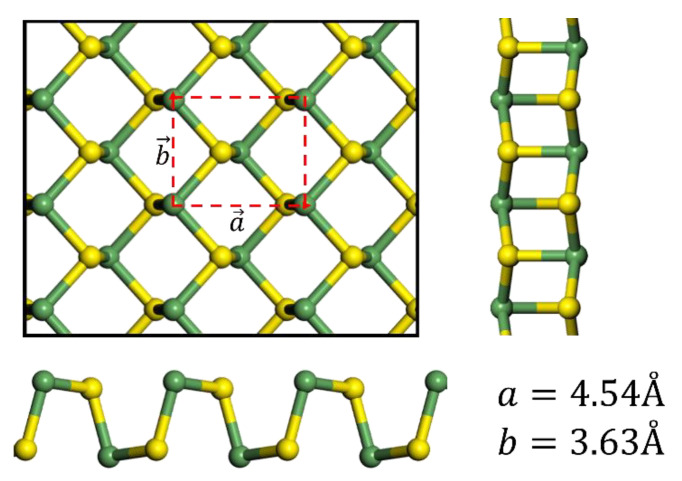
Structure and lattice parameters of the pristine GeS layer. Germanium and sulfur atoms are represented by green and yellow balls, respectively. The 1 × 1 unit cell is marked with the red box.

**Figure 2 nanomaterials-11-02960-f002:**
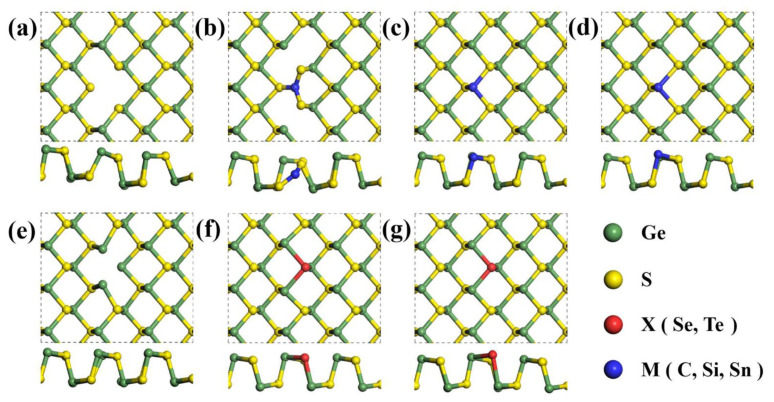
Structures of (**a**) Ge vacancy ($V_{Ge}$) and (**b**) substitutional C ($C_{Ge}$), (**c**) substitutional Si ($Si_{Ge}$), and (**d**) substitutional Sn ($Sn_{Ge}$) for Ge vacancy; structures of (**e**) S vacancy ($V_{S}$) and (**f**) substitutional Se ($Se_{S}$) and (**g**) substitutional Te ($Te_{S}$) for S vacancy. For perspective views of atomic structures of pristine and defective GeS are shown in Appendix A.

**Figure 3 nanomaterials-11-02960-f003:**
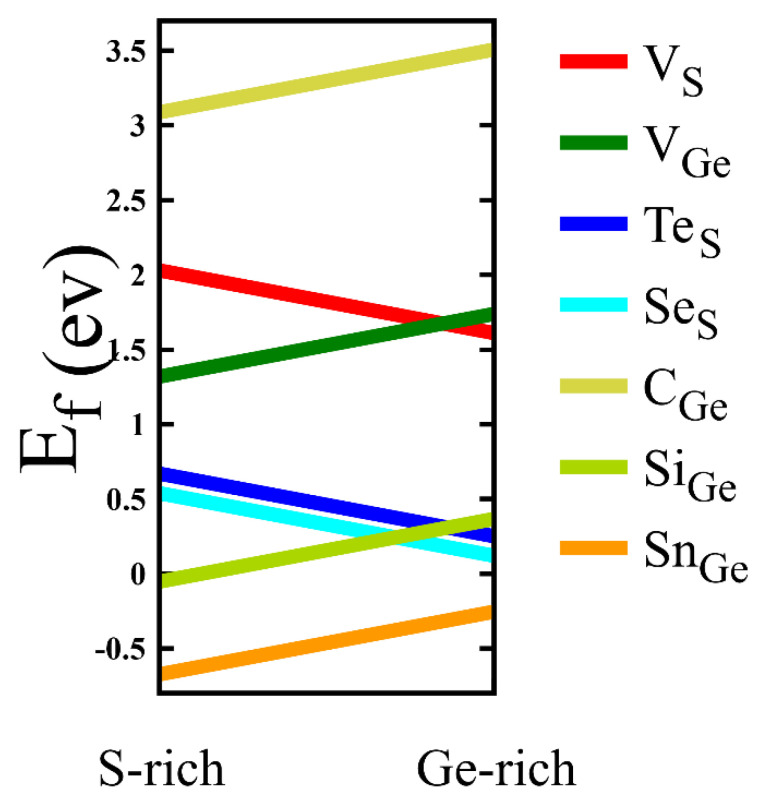
Formation energies of defected structures as a function of chemical potential. Especially, the formation energy of $Sn_{Ge}$ is negative for S-rich and Ge-rich conditions, which implies that $Sn_{Ge}$ is energetically favorable.

**Figure 4 nanomaterials-11-02960-f004:**
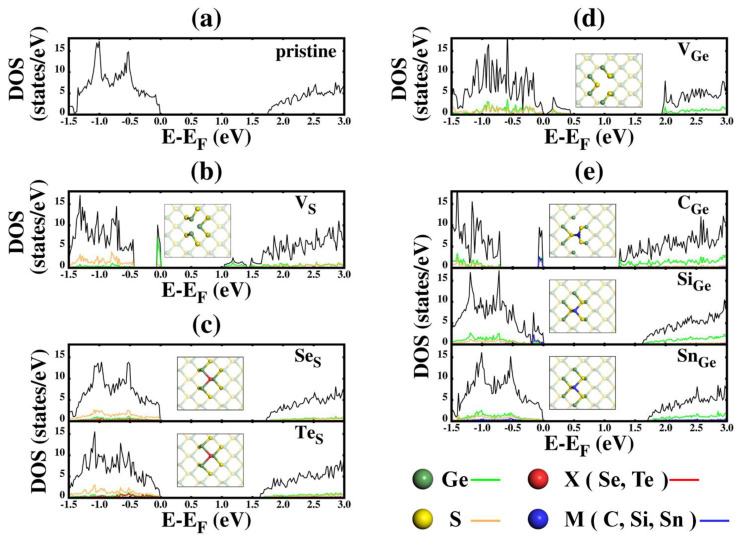
Total DOS and PDOS of monolayer pristine GeS and defective GeS: (**a**) pristine GeS, (**b**) GeS with vacancy at S site $\left(V_{S}\right)$, (**c**) GeS with substitutional atoms at S site (${\mathrm{Se}}_{S}{\;\mathrm{and}\;\mathrm{Te}}_{S}),$ (**d**) GeS with vacancy at Ge site $\left(V_{\mathrm{Ge}}\right),$ and (**e**) GeS with substitutional atoms at Ge site $\left(C_{\mathrm{Ge}},{\mathrm{Si}}_{\mathrm{Ge}},{\mathrm{Sn}}_{\mathrm{Ge}}\right)$. The black line represents total DOS, while the colored PDOS lines represents contributions by the atoms shown in the same color in the inset.

**Table 1 nanomaterials-11-02960-t001:** Bandgaps obtained from DFT calculations for GeS without and with defect.

	Pristine GeS	$V_{Ge}$	$V_{S}$	$C_{Ge}$	$Si_{Ge}$	$Sn_{Ge}$	$Se_{S}$	$Te_{S}$	Pristine GeS DFT [75]/Exp [76,77]
$E_{G}$ (eV)	1.76	0.04	1.04	1.26	1.60	1.71	1.71	1.63	1.65/1.65, 1.70–1.96

**Table 2 nanomaterials-11-02960-t002:** Elastic coefficients for pristine GeS, GeS with vacancies ($V_{\mathrm{Ge}}$, $V_{S}$), and GeS with substitutional atoms (${\mathrm{Sn}}_{\mathrm{Ge}}$, ${\mathrm{Se}}_{S}$) compared with previous studies for pristine GeS and ${\mathrm{MoS}}_{2}$.

	Pristine GeS	$V_{Ge}$	$V_{S}$	$Sn_{Ge}$	$Se_{S}$	Pristine GeS [33]/[34]	$MoS_{2}$
$C_{11}\left(N/m\right)$	13.28	11.01	11.84	14.15	13.95	15.24/20.87	129.94/130 [53]
$C_{22}\left(N/m\right)$	44.28	42.74	43.76	48.87	49.16	45.83/53.40	130.57/130 [53]
$C_{12}\left(N/m\right)$	18.71	12.13	16.11	19.09	19.26	21.62/22.22	32.03/32 [53]

**Table 3 nanomaterials-11-02960-t003:** Piezoelectric stress coefficients $e_{ij}$ for pristine GeS, GeS with vacancies ($V_{\mathrm{Ge}}$, $V_{S}$), and GeS with substitutional atoms (${\mathrm{Sn}}_{\mathrm{Ge}}$, ${\mathrm{Se}}_{S}$) compared with previous studies for pristine GeS and ${\mathrm{MoS}}_{2}$.

	Pristine GeS	$V_{Ge}$	$V_{S}$	$Sn_{Ge}$	$Se_{S}$	Pristine GeS [33]/[34]	$MoS_{2}$
$e_{11}\left(C/Å\right)$	5.84	7.92	5.72	5.94	5.83	7.28/4.6	3.66/3.64 [53]
$e_{12}\left(C/Å\right)$	−4.59	−8.29	−3.67	−4.36	−4.51	−4.97/−10.1	−3.67/−3.64 [53]

**Table 4 nanomaterials-11-02960-t004:** Piezoelectric strain coefficients $d_{ij}$ for pristine GeS, GeS with vacancies ($V_{\mathrm{Ge}}$, $V_{S}$), and GeS with substitutional atoms (${\mathrm{Sn}}_{\mathrm{Ge}}$, ${\mathrm{Se}}_{S}$) compared with previous studies for pristine GeS and MoS_2_.

	Pristine GeS	$V_{Ge}$	$V_{S}$	$Sn_{Ge}$	$Se_{S}$	Pristine GeS [33]/[34]	$MoS_{2}$
$d_{11}\left(pm/V\right)$	144.71	135.75	119.66	114.19	118.62	190.92/75.43	3.74/3.73 [53]
$d_{12}\left(pm/V\right)$	−71.51	−57.92	−52.44	−53.53	−55.64	−100.91/−50.42	−3.73/−3.73 [53]

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
