# Peer review of "Effect of Point Defects on Electronic Structure of Monolayer GeS"

_nanomaterials, 2021, doi:10.3390/nano11112960_

Round 1

Reviewer 1 Report

The manuscript reports a density functional theory study of point defects in monolayer GeS using a supercell approach. The authors employ state-of-the-art computational methodology, and the results are useful enough to be published in Nanomaterials. However, before the paper can be published, the authors should address the following points:

1. The authors should better explain the concept of the total energy per formula unit they use to derive the formation energy: It is not immediately evident that the chemical potentials of the individual atoms can be calculated as the total energies of the specific bulk crystals. A justification of this approach should be presented and the relevant literature should be cited.

2. The density of states curves in Figure 4 are presented in arbitrary units. This is not sufficient. The authors should revise the figure to show the actual units.

Reviewer 2 Report

The authors detailedly investigated the effect of vacancies and substitutional atoms on electronic structure of monolayer GeS using density functional theory.
Authors results show that the bandgap of GeS with substitutional atoms is close to that of pristine GeS, while the bandgap of GeS with Ge or S vacancies is smaller than that of pristine GeS. In addition, defects affect the piezoelectric properties depending on vacancies or substitutional atoms. 
Especially, GeS with substitutional atoms has almost the same piezoelectric stress coefficients e ij as pristine GeS, while having lower piezoelectric strain coefficients dij but still much higher 7 than other 2D materials.
The present manuscript reports is interesting. Prior to publication the authors should address the following questions.

1- It seems that lines 88-97 and 100-109 of manuscript were repeated with no reason.
2- Authors have substituted different atoms with Ge and S. They should explain what are the main results of substituting Ge atom by C atom (CGe).
3- Authors should explain clearly the utility or deficiency of defects on piezoelectric effects of GeS. 
4- It seems after introducing vacancies and dopants, a deformation occurs in the crystal structure! This should be shown by cross-sectional views in Fig.2 and also given lattice constants in a Table.
5- Do the authors have any idea on the spin polarization of the compound? If no spin polarization is expected, then please explain it in brief.
6- The authors must explain whether the analyzed structures are direct or indirect semiconductors. Also, the location of the VBM and CBM should be mentioned.
7- To have an insight on the transport mechanism of the structures, the charge transfer analysis is necessary.

8- DFT calculations for 2D materials are a very lively field. In general, one should cite more relevant previous works about defects and substitutional atoms and strains on the 2D monolayers. (Refer to : Appl. Surf. Sci. 559, 149862; J. Phys. Chem. C 125, 13067; J. Appl. Phys. 129, 155103;Sci. Rep. 10 (1), 1-15;Phys. Chem. Chem. Phys. 23, 21196;Phys. Chem. Chem. Phys. 23, 15319;Phys. Chem. Chem. Phys. 23, 15216)    

Reviewer 3 Report

The electronic structure of monolayer GeS is modeled in this work using the DFT-based generalized gradient approximation (GGA) within the Perdew-Burke-Ernzerhof (PBE) form in the Vienna ab initio simulation package (VASP). For the considered GeS monolayer, piezoelectric  properties and elastic coefficients are also calculated by the density functional  perturbation theory (DFPT) method. The stability of the GeS monolayer with defects is also estimated. 

The similarity of GeS and other monolayered compounds is clear. The research is well-organized and presented in detail. Some parts are missing. The manuscript can be revised to include the following changes:

  1. In fact, I found no appropriate references and discussions of GeS in Introduction. This should be corrected. There are some papers on GeS published in the literature. 
  2. In Tables 2 and 3 only [33] is taken for comparison. Definitely, there are other papers published on GeS. I recommend to cite them.  
  3. The band gap value is not compared to any other calculation and experimental results. Eg=1.65eV and higher values can be found in the literature, e.g., Eymard, R., Otto, A.: Phys. Rev. B 16 (1977) 1616. 
  4. Figure 3 is hard to comprehend. The numbers are tiny, the width should be increased. Was it calculated for two points for each line? This can be discussed in the text.
  5. A band structure plot is missing in this manuscript. For pristine GeS, it is required to determine a direct or indirect character of the band gap. The corresponding k-points or directions are important for any further study.  
